# Comprehensive Genetic and Molecular Characterization Confirms Hepatic Stellate Cell Origin of the Immortal Col-GFP HSC Line

**DOI:** 10.3390/ijms26167764

**Published:** 2025-08-11

**Authors:** Larissa F. Buitkamp, Thomas Liehr, Stefanie Kankel, Eva M. Buhl, Katharina S. Hardt, Diandra T. Keller, Sarah K. Schröder-Lange, Ralf Weiskirchen

**Affiliations:** 1Institute of Molecular Pathobiochemistry, Experimental Gene Therapy and Clinical Chemistry (IFMPEGKC), RWTH University Hospital Aachen, D-52074 Aachen, Germany; larissa.buitkamp@rwth-aachen.de (L.F.B.); khardt@ukaachen.de (K.S.H.); dikeller@ukaachen.de (D.T.K.); 2Jena University Hospital, Institute of Human Genetics, Friedrich Schiller University, D-07747 Jena, Germany; thomas.liehr@med.uni-jena.de (T.L.); stefanie.kankel@med.uni-jena.de (S.K.); 3Electron Microscopy Facility, Institute of Pathology, RWTH University Hospital Aachen, D-52074 Aachen, Germany; ebuhl@ukaachen.de

**Keywords:** liver, hepatic stellate cell(s), cell authentication, STR profile, karyogram, M-FISH analysis, Cellosaurus, International Cell Line Authentication Committee (ICLAC)

## Abstract

The immortal murine hepatic stellate cell line Col-GFP HSC was comprehensively characterized using genetic and molecular approaches. Short tandem repeat (STR) profiling and karyotyping combined with multiplex fluorescence in situ hybridization (M-FISH) confirmed the identity of the cell line and revealed no contamination. Col-GFP HSCs showed a near tetraploid karyotype. Additionally, next-generation sequencing (NGS) data, quantitative reverse transcription PCR, and Western blot analyses demonstrated robust expression of genes and proteins associated with hepatic stellate cells, including those involved in extracellular matrix remodeling and fibrogenic pathways. Phalloidin staining revealed filamentous actin patterns characteristic of stellate cells, providing additional support for their cytoskeletal organization and functional status. These findings provide strong evidence that the Col-GFP HSC cell line originates from hepatic stellate cells and can serve as a reliable in vitro model to study stellate cell biology and related pathophysiological processes.

## 1. Introduction

Hepatic stellate cells (HSCs) play a crucial role in maintaining liver homeostasis and are essential to the development of hepatic fibrosis [1]. In their quiescent state, HSCs store vitamin A and help regulate normal liver architecture. However, when the liver is injured, these cells activate into a myofibroblast-like phenotype and contribute to the excessive deposition of extracellular matrix (ECM) [1]. This process is central to the pathogenesis of chronic liver diseases, including cirrhosis, highlighting the importance of HSCs as therapeutic targets for mitigating liver damage. Nevertheless, the use of primary HSCs in research is often hindered by donor variability, low proliferation rates, and premature senescence in culture, factors that can complicate detailed mechanistic studies and drug screening efforts [2].

To overcome these challenges, immortalized HSC lines such as Col-GFP HSCs have been developed, providing readily available model systems that maintain critical stellate cell features. In this paper, we present a comprehensive genetic and molecular evaluation of Col-GFP HSCs. We incorporate short tandem repeat (STR) profiling and karyotyping combined with multiplex fluorescence in situ hybridization (M-FISH) to verify cellular identity and stability, determine ploidy, and identify possible copy number variations. We also examine gene and protein expression using next-generation sequencing (NGS), quantitative real-time PCR (qRT-PCR), and Western blot analyses, as well as phalloidin staining to reveal the cytoskeletal attributes characteristic of stellate cells. Taken together, these methods robustly authenticate the Col-GFP HSC line, emphasizing its value as a reliable in vitro model for studying HSC biology and fibrogenic processes.

Finally, it is crucial to emphasize the importance of using authenticated and genetically characterized cell lines in biomedical research. Relying on misidentified or cross-contaminated cell lines can lead to inconsistent results, wasted resources, and compromised project outcomes [3]. Verifying the genetic identity, stability, and lineage of cell models strengthens the reproducibility and reliability of their data, fostering higher scientific rigor standards. The Col-GFP HSC cell line is a prime example of how robust genetic and molecular characterization can promote confidence in experimental findings and support translational advancements in hepatic research.

Established in 2013, the Col-GFP HSC cell line has so far only been reported in ten publications, indicating its limited but growing recognition within the scientific community (Table 1).

To enhance accessibility for researchers, this cell line has recently been deposited with the non-profit DSMZ (German Collection of Microorganisms and Cell Cultures GmbH) under the accession number ACC 941, ensuring its availability for future studies. Recognizing the significance of comprehensive characterization, we have expanded our genetic and transcriptomic analyses in the present study to provide researchers with robust data to assist them in determining if this specific HSC model is suitable for their experimental requirements.

## 2. Results

### 2.1. Microscopic Appearance of the Col-GFP HSC Line

Microscopic analysis consistently reveals that the Col-GFP HSC line has a fibrogenic appearance. This is confirmed by well-developed cytoskeletal structures and stress fibers reminiscent of those found in myofibroblast-like HSCs (Figure 1). At lower cell densities, they form short filopodia, whereas as the culture becomes denser, the cells grow irregularly and begin to overlap (Figure 1A). The hallmark feature of this particular cell line is its capacity to express green fluorescent protein (GFP) under the control of collagen regulatory elements, facilitating the direct visualization and quantification of collagen activity in real time (Figure 1B). Consequently, elevated GFP fluorescence correlates with intensified fibrogenesis, providing a simple, non-invasive reporter system for monitoring the fibrogenic state and collagen production under different experimental conditions. For comparison, the murine AML12 hepatocyte cell line, which exhibits more irregular growth in culture, is depicted. Their filopodia are visibly more pronounced than those of Col-GFP HSC cells. Moreover, when comparing the morphology of Col-GFP HSC with other immortalized hepatic stellate cell lines (GRX, CFSC-2G, HSC-T6, PAV-1, and LX-2) or other fibrogenic cell lines (Rat-1 and NuFF), it is evident that each cell line exhibits unique morphological features (Figure 1C). For instance, HSC-T6, PAV-1, Rat-1, and NuFF cells appear more spindle-shaped, while Col-GFP HSC, CFSC-2G, and LX-2 cells display more polygonal configurations. Additionally, CFSC-2G and LX-2 have a flatter contour, whereas HSC-T6 and PAV-1 form more elongated filopodia reminiscent of active HSCs. Among the fibroblastic cell lines, NuFF cells have a more uniform, flattened morphology compared to Rat-1 cells. However, while morphological traits such as spindle-shaped or polygonal forms, varying degrees of stress fiber development, and filopodial extensions are useful for distinguishing different hepatic stellate and fibroblastic cell lines. Because the Col-GFP reporter is stably integrated into Col-GFP HSCs, GFP fluorescence provides a convenient, albeit artificial, read-out for collagen I promoter activity in this HSC line.

### 2.2. Actin Cytoskeleton Organization in the Col-GFP HSC Line

Rhodamine-phalloidin staining revealed prominent filamentous actin networks in the Col-GFP HSC line (Figure 2A), highlighting myofibroblast-like characteristics and emphasizing the fibrogenic nature of these cells. In comparison, AML12 cells exhibit a distinct actin arrangement with fewer thick stress fibers that are more diffuse and cortical compared to the pronounced, myofibroblast-like actin bundles observed in Col-GFP HSC (Figure 2B).

### 2.3. Electron Microscopic Analysis of Col-GFP HSCs

HSCs exhibit distinctive features when assessed through electron microscopy [14,15,16,17,18]. In particular, these cells are characterized by their long cytoplasmic processes, facilitating communication and interaction within the liver microenvironment. A prominent feature of HSCs is the presence of numerous lipid droplets (LDs) in their cytoplasm, which serve as a storage for vitamin A and other lipophilic substances, highlighting their role in metabolic functions. Additionally, HSCs contain a well-developed rough endoplasmic reticulum (ER), indicating their active involvement in protein synthesis, particularly during liver injury and fibrosis. The nuclei of hepatic stellate cells are often large and irregularly shaped, reflecting their metabolic activity and potential for proliferation in response to liver damage [14,15,16,17,18].

To verify these features, we performed electron microscopy and observed some of these unique morphological traits in Col-GFP HSC, including extensive rough ER indicative of a high capacity for protein synthesis, prominent stress fibers associated with a myofibroblast-like state, and occasional lipid droplets reflecting the functionality of stellate cells. These characteristic subcellular structures further affirm that the Col-GFP HSC line shares essential morphological hallmarks with primary HSCs (Figure 3).

### 2.4. Short Tandem Repeat Profiling of the Col-GFP HSC Line

The authenticity of the Col-GFP HSC cell line was confirmed using STR profiling, as previously published [4]. A comparison of the STR profile using the Cell Line Authentication using STR (CLASTR) revealed that the Col-GFP HSC line has a unique STR profile (Appendix A) and similarities to profiles originating from C57BL/6 mice, confirming that the Col-GFP HSC cell line is derived from a C57BL/6 genetic background (Appendix A). Additionally, we profiled the murine AML12 cell line used as a control cell originating from hepatocytes in our study. Our analysis showed that our line had an 88.8% identical profile (seven or eight markers were identical) to the profile deposited for this cell line in Cellosaurus [19] (Appendix A).

### 2.5. Karyotype Analysis Combined with Multiplex Fluorescence in Situ Hybridization (M-FISH) of Col-GFP HSCs

Karyotype analysis and M-FISH were performed on Col-GFP HSCs to evaluate chromosomal stability and confirm their murine origin (Figure 4). Inverted 4′,6-diamidino-2-phenylindole (DAPI)-banding revealed a relatively stable set of chromosomes, with structural variations typical of an immortalized cell line, and no signs of contamination or cross-species contamination (Figure 4A). M-FISH analysis further confirmed these results by color-coding each chromosome and displaying characteristics of a Black6-derived cell line (Figure 4B). Combined, these cytogenetic methods indicated that the Col-GFP HSC cell line had a near-tetraploid karyotype, described as: 66~71<4n>,XXY,-1,-3,der(4)t(4;6)(E;A2),-4,-6,-7,-9,-12,-13,-14,-15,-18 [17]/67~72<4n>,XXYY,idem [6]/73<4n>,XXYYY,idem [1].

More specifically, we found that the Col-GFP HSC cell line exhibits several distinctive features that can be used for authentication purposes. Our karyotyping revealed that Col-GFP HSCs have a near-tetraploid chromosome complement with one notable structural anomaly and several numerical anomalies. The predominant karyotype (66–71 chromosomes, XXY) showed the loss of several chromosomes, including chromosomes 1, 3, 4, 6, 7, 9, 12, 13, 14, 15, and 18, as well as a derivative chromosome 4 resulting from a translocation with chromosome 6. Some cells exhibited minor variations in the form of additional sex chromosomes (e.g., XXYY or XXYYY). Additionally, rather than having chromosome numbers between 66 and 73, one cell exhibited near-octoploid conditions with 121 or 140 chromosomes per cell. This cell was not characterized in detail. However, these variations appear to be due to endomitosis of typical near-tetraploid Col-GFP HSC cells. When we compared the observed genotype to that expected from a mouse strain such as C57BL/6, significant differences became apparent. C57BL/6-derived cells would display a normal diploid (2n) karyotype of 40 chromosomes, without the translocations, tetraploidization, or multiple sex chromosomes observed here. Overall, the observed aberrations in Col-GFP HSCs are consistent with those of an immortalized C57BL/6 cell line that has adapted to in vitro culture since its establishment around ten years ago.

### 2.6. Next Generation mRNA Sequencing of Col-GFP HSC

We conducted NGS analysis on Col-GFP HSC cells cultured under standard conditions to gain a deeper understanding of their transcriptional landscape. As expected for a fibrogenic HSC line, the resulting expression profile revealed the expression of key genes associated with myofibroblast-like features and ECM remodeling. These genes included collagen isoforms (*Col1a1*, *Col1a2,* and *Col1a3*), α-smooth muscle actin (*Acta2*), tissue inhibitor of metalloproteinases 1 (*Timp1*), decorin (*Dcn*), vinculin (*Vln*), transgelin (*Tagln*), insulin-like growth factor binding proteins (*Igfbp6*, *Igfbp7*), secreted acidic cysteine-rich glycoprotein (*Sparc*), cellular communication network factors (*Ccn1*/*Cyr61*, *Ccn2/Ctgf*), cysteine and glycine-rich protein 2 (*Csrp2*), and others (Appendix A). We also expected the expression of genes associated with transforming growth factor-β (TGF-β) signaling, such as the TGF-β isoforms (*Tgfb1*, *Tgfb2*, and *Tgfb3*), relevant TGF-β receptors or co-receptors (*Tgfbr1*, *Tgfrb2*, and *Tgfrb3, Eng*), binding proteins (*Ltbp1*-*Ltpb4*), and genes induced by TGF-β, which play a central role in HSC activation and fibrogenesis. However, we found low or no expression of desmin (*Des*), synaptophysin (*Syp*), reelin (*Reln*), fibroblast growth factor 10 (*Fgf10*), or glial fibrillary acidic protein (*Gfap*), which are described as markers for HSCs in some studies [20,21,22,23,24].

The NGS analysis provided a detailed transcriptional profile, revealing that genes typically associated with other hepatic cell types, such as hepatocytes, Kupffer cells, liver sinusoidal endothelial cells (LSECs), and cholangiocytes, are either not expressed or present only at very low levels in Col-GFP HSCs. For example, genes central to hepatocyte function, such as albumin (*Alb*), hepatocyte nuclear factor 4 alpha (*Hnf4a*), and alpha fetoprotein (*Afp*), were scarcely detected or absent (Appendix A).

Similarly, Kupffer cell markers including the monocyte differentiation antigen *Cd14,* the leukocyte activation antigen *Cd38*, the myeloperoxidase *(Mpo),* and the adhesion G protein-coupled receptor E1 *Adgre1* (F4/80) showed minimal or no expression, indicating a lack of macrophage-like features (Appendix A). Moreover, endothelial markers like *Pecam-1* (CD31) and von Willebrand factor (*Vwf*) that characterize LSEC were substantially downregulated or absent (Appendix A). Genes typically expressed in cholangiocytes, such as trefoil factors (*Tff1*, *Tff2*, *Tff3*), the epithelial cellular adhesion molecule (*Epcam*), or mucin 5, subtype B, tracheobronchial (*Muc5b*), were undetectable or showed low expression in Col-GFP HSCs (Appendix A). The notable exception was keratin 19 (*Krt19*), which was moderately expressed.

The absence of these lineage-specific transcripts highlights that Col-GFP HSCs do not exhibit characteristics typical of other hepatic cell subclasses. In contrast, the cell line exhibited a transcriptional signature more consistent with activated stellate cells. This supports the conclusion that the Col-GFP HSC cell line indeed derives from HSCs, rather than from another parenchymal or non-parenchymal liver lineage.

### 2.7. Validation of Expression Data Using Conventional and Quantitative Real-Time PCR

To validate our NGS results at the transcriptional level, we performed conventional RT-PCR targeting genes characteristically expressed in HSCs and hepatocytes, or found to be undetectable (*Alb*, *Foxa2*, *Hnf4a*, and *Rn18S*) or low expressed (*Acta2*) at the mRNA level in Col-GFP HSCs (Figure 5A). Additionally, we conducted several real-time RT-qPCRs for genes associated with hepatocytes or HSCs, targeting well-established markers indicative of their respective cellular identities, to further confirm our NGS data (Figure 5B,C). In all these experiments, gene expression patterns in Col-GFP HSCs were compared with those observed in the AML12 murine hepatocyte line.

Consistent with the NGS sequencing data, Col-GFP HSCs exhibited expression of stellate cell-associated genes, while AML12 cells displayed high levels of hepatocyte-specific transcripts. Moreover, several markers specific for hepatocytes, Kupffer cells, liver sinusoidal endothelial cells, and cholangiocytes showed low or no expression in Col-GFP HSC cells. These findings confirm that Col-GFP HSCs maintain an HSC phenotype, thereby reinforcing the conclusions drawn from the NGS analysis.

RT-qPCR analysis revealed significant differences in gene expression between Col-GFP HSC and AML12 cells. The relative mRNA expression was significantly higher in AML12 cells compared to Col-GFP HSC for the following genes: ferritin heavy chain 1 (*Fth1*), hepatocyte nuclear factor 4 alpha (*Hnf4ɑ*), cyclin D1 (*Ccnd1*), actin related protein 2/3 complex subunit 2 (*Arpc2*), perilipin 2 (*Plin2*), C-C- motif chemokine ligand 2 (*Ccl2*), C-X-C motif chemokine ligand 5 (*Cxcl5*), and solute carrier family 40 member 1 (*Slc40a1*). Conversely, the relative mRNA expression was higher for proliferating cell nuclear antigen (*Pcna*), cyclophilin A (*Ppia*), lysosomal-associated membrane protein 1 (*Lamp1*), glutamic-oxaloacetic transaminase 2 (*Got2*), ribosomal protein S6 (*Rps6*), lysosomal-associated membrane protein 2 (*Lamp2*), voltage-dependent anion channel 1 (*Vdac1*), and triosephosphate isomerase 1 (*Tpi1*) in Col-GFP HSCs. No significant differences in relative mRNA expression were detected for the genes ferritin light chain 1 (*Ftl1*), actinin alpha 1 (*Actn1*), vimentin (*Vim*), β-actin (*Actb*), 18S ribosomal RNA (*Rn18s-rs5*), ribosomal protein L13a (*Rpl13a*), high mobility group box 1 (*Hmgb1*), tropomyosin 1 (*Tpm1*), and peroxiredoxin 1 (*Prdx1*) between the two cell lines.

### 2.8. Validation of Expression Data by Western Blot Analysis

To confirm the NGS findings at the protein level, we conducted Western blot analyses using a panel of markers expected to be expressed in Col-GFP HSC cells based on the NGS data. Murine AML12 hepatocyte cells and mouse liver tissue served as controls. The markers chosen are known to be (more or less) specific for HSCs and include fibronectin and vimentin. Both markers were prominently detected in Col-GFP HSCs (Figure 6). Additionally, we verified the presence of the SV40T antigen and the green fluorescent protein (GFP), both of which are part of the transgene used to immortalize this cell line. We also evaluated the expression of typical hepatocyte markers that were either absent at the RNA level, such as hepatocyte nuclear factor 4α (HNF4α), or expressed at low levels, such as albumin, in Col-GFP HSCs. In contrast to AML12, which exhibited strong signals for HNF4α, Col-GFP HSCs did not show expression of this hepatocyte marker, confirming our NGS findings. Furthermore, both AML12 and Col-GFP HSC cells did not express albumin at the protein level, but displayed high levels of general markers such as actinin 1, heat shock protein 90 (HSP90), proliferating cell nuclear antigen (PCNA), ferritin light polypeptide 1 (FTL1), and ferritin heavy polypeptide 1 (FTH1), while both cell lines tested positive for Caveolin 1. Interestingly, Cyclin D1 expression was not detected in Col-GFP HSC cells, despite showing weak expression at the mRNA level (TPM: 13.69; Appendix A).

### 2.9. Comparison of Marker Expression in Hepatic Stellate and Fibroblastic Cell Lines

To better understand the fibrogenic and myofibroblast-like characteristics of Col-GFP HSCs compared to various other HSC and fibroblastic cell lines, we examined the expression of key markers involved in extracellular matrix production and cytoskeletal organization. These markers, including collagen types I and IV, α-smooth muscle actin (α-SMA), fibronectin, and vimentin, are commonly used to assess the degree of cellular activation, fibrogenesis, and overall phenotype. Our analysis reveals distinct expression profiles across the HSC and fibroblastic cell lines tested. While α-SMA, fibronectin, and vimentin are present in all cells at varying levels, collagen type I is only detectable in CFSC-2G, PAV-1, Rat-1, WI-38, and NuFF (Figure 7). In contrast, collagen type IV expression is restricted to the human cell lines LX-2, WI-38, and NuFF, while GAPDH, used as a control for equal protein loading, was consistently expressed in all samples. However, it should be noted that the proteins tested were of mouse, rat, or human origin, possibly showing variant immunoreactivity against the chosen antibodies. Moreover, testing with an antibody directed against the large T antigen of SV40 confirmed that Col-GFP HSCs are positive for this oncogene used for immortalization, similar to the WI-38 cell line, representing a lung fibroblastic cell line that was also immortalized by transfection with this oncogene.

## 3. Discussion

HSCs play a central role in hepatic fibrosis, primarily due to their ability to produce large amounts of extracellular matrix [10]. While primary HSCs offer valuable insights into fibrogenic processes, they have a limited lifespan in culture and are difficult to isolate in sufficient quantity. Ethical regulations further restrict the use of animals in research. As a result, many laboratories rely on immortalized HSC lines, which can be generated through various methods such as transduction with the Simian virus 40 (SV40) large T-antigen, introduction of telomerase reverse transcriptase (TERT) activity, or spontaneous immortalization in diseased livers [10]. These immortalized lines provide a practical, ethically sound, and reproducible model system that aligns with the 3R principle of reducing, refining, and replacing animal models [10].

In terms of their phenotype, quiescent HSCs often contain lipid droplets enriched in retinoids and display a notable Golgi apparatus, as well as a developed endoplasmic reticulum [14,15,16,17,18]. When activated, they transdifferentiate into myofibroblastic cells and exhibit denser cytoplasmic microfilaments. Classic HSC markers include glial fibrillary acidic protein (GFAP) [23,24], synaptophysin [20], and several intermediate filament proteins such as desmin [25], nestin [26,27], and vimentin [9,28]. However, α-SMA and collagen levels typically increase upon activation. Despite these shared characteristics, the different methodologies used to generate immortalized lines can lead to significant variability in their marker gene expression. Some SV40 large T-antigen-immortalized lines, for example, may exhibit altered regulation of filament proteins compared to spontaneously immortalized lines, which can arise from cirrhotic or otherwise diseased livers [10]. These differences underscore the necessity of thorough authentication, including short tandem repeat profiling and careful phenotypic assessment, when selecting an immortalized HSC line for any given experiment. Such verification helps ensure accurate interpretation of results and preserves consistency across different research labs, given that HSC lines, despite their convenience and utility, may not perfectly replicate the properties of primary HSCs.

Moreover, cell tracking analyses and single-cell RNA sequencing have revealed a previously underappreciated heterogeneity within HSCs and their activated derivatives, known as myofibroblasts. These analyses demonstrated that populations of HSCs can follow distinct activation pathways under normal and injured states [29,30,31,32,33]. This finding emphasizes the complexity of HSC biology and underscores the importance of carefully selecting cell lines or subsets for specific experimental goals.

This study provides a comprehensive genetic and molecular characterization of the immortal Col-GFP HSC cell line, reinforcing its identity as an HSC model. However, certain observations warrant careful consideration. Extensive mRNA and protein expression data revealed that Col-GFP HSCs lack significant expression of the signatures typically associated with other liver cell types. These include hepatocyte-specific Albumin and Hnf4α, Kupffer cell markers CD68 and F4/80, and endothelial markers CD31 and von Willebrand factor (vWF). Instead, the cell line displayed expression of *Acta2* (α-smooth muscle actin), collagen isoforms, and other fibrogenic markers, which align well with the established profile of activated stellate cells involved in ECM remodeling. Microscopically, the presence of abundant stress fibres and the ability to express GFP under the control of collagen regulatory elements provides a direct and convenient readout of fibrogenic activity. Meanwhile, electron microscopy confirmed morphological traits such as extensive rough endoplasmic reticulum and occasional lipid droplets, underscoring an HSC-like ultrastructure.

Despite compelling evidence supporting the HSC origin, certain observations diverge from the idealized stellate cell phenotype. The most notable of these is the moderate expression of keratin 19 (KRT19), which is commonly recognized as a cholangiocyte marker, but is occasionally detected at low levels in activated stellate cells, in hepatic progenitor cells, or under in vitro conditions that permit partial transdifferentiation [34,35,36]. Another notable point is the cytogenetic data, which indicate near-tetraploidy and the presence of additional sex chromosomes, as well as other structural rearrangements. While these anomalies may affect certain pathways, they do not invalidate the use of Col-GFP HSCs as a model, provided that researchers are mindful of how these aberrations might influence experimental results.

Nevertheless, out of the 46 proteins listed in the PanaoDB database as HSC markers [37], 33 markers (71.7%) were expressed at different quantities at the mRNA level. These markers include *Cygb*, *Pparg*, *Pdgfra*, *Foxf1*, *Alb*, *Nr1h4*, *Ctgf/Ccn2*, *Agtr1a, Wt1*, *Vegfa*, *Acta2*, *Myb*, *Des*, *Vcl*, *Timp1*, *Col1a1*, *Tagln*, *Col1a2*, *Col3a1*, *Sparc*, *Rbp1*, *Ddn*, *Myl9*, *Tpm2*, *Meg3*, *Bgn*, *Igfbp7*, *Igfbp3*, *Cyr61/Ccn1*, *Olfml3*, *Igfbp6*, *Ccl2*, and *Colec11*). However, 13 markers (*Adamts13*, *Rgs5*, *Ppara*, *Sema7a*, *Pnpla3*, *Fgf10*, *Gfap*, *Ngfr*, *Fap*, *Slc8a1*, *Reln*, *Syp*, and *Hgf*) were not detectable. Some of these markers commonly considered classical for HSCs are useful in distinguishing HSCs from other liver cell types [38]. However, our study has revealed that these markers can also be expressed in non-related fibrogenic cells, indicating that they may not be reliable indicators of HSCs. This is especially true for α-SMA, fibronectin (*Fn1*), and vimentin (*Vim*), which were also detected at the protein level in Rat-1, WI-38, and NuFF cells. Additionally, we found that *Gfap*, *Des*, lecithin-retinol acyltransferase (*Lrat*), and retinol binding protein 1 (*Rbp1*), which are typically selective for quiescent/early-activated HSCs, were expressed at very low levels or were virtually absent in Col-GFP HSCs. In contrast, *Timp1*, platelet-derived growth factor-β (*Pdgfrb*), and lysyl oxidase-like 2 (*Loxl2*) were expressed in higher quantities (Appendix A). This suggests that the phenotype of Col-GFP HSCs more closely resembles that of transdifferentiated, extracellular matrix-producing myofibroblast cells rather than early-activated HSCs.

A key aspect of this work is the rigorous cell authentication strategy. By performing STR profiling and comparing the results with those of previously published C57BL/6 reference profiles, we can confirm the murine origin of the cells and rule out cross-species contamination. This mitigates one of the major risks in in vitro research, which is the use of misidentified or contaminated cell lines. Such precautions are essential for ensuring data reproducibility and reliability [3].

The near-tetraploid karyotype and derivative chromosomes, as revealed through karyotyping and M-FISH analyses, highlight the importance of reporting and monitoring chromosomal integrity in immortalized models over time. Tetraploidization is common in immortal cell lines and usually occurs within the first few years of establishment. Chromosomal instability is more commonly observed in tumor cell lines. Therefore, the loss of over ten chromosomes within approximately 12 years of establishing the cell line is somewhat surprising. However, the loss of one Y chromosome in the main clone is consistent with gonosomal loss in murine cell lines, specifically the loss of Y chromosomes, which has been observed in 17 out of 26 murine (tumor) cell lines [39]. Overall, Col-GFP HSCs should be monitored for further acquired changes over the next few decades to prevent them from being studied as a tumor-like immortal cell line rather than as a model for HSCs.

From a functional perspective, the markers used in this study play a crucial role in validating HSC identity. *Acta2*, which encodes α-SMA, was weakly expressed in our NGS analysis, confirming our previous report showing that this activation marker is only expressed at low levels [4]. Likewise, the NGS data showed that the neurogenic crest cell marker GFAP is expressed in low mRNA quantities, confirming our previous report [4]. Although this marker is expressed at low mRNA levels, we demonstrated in our previous study that this protein can be detected by immunofluorescence staining. Additionally, other proteins reflecting the myofibroblast-like properties of Col-GFP HSCs, such as collagen genes and tissue inhibitors of metalloproteinases (TIMPs), confirmed the capacity for ECM production [1]. Conversely, the absence or greatly reduced expression of albumin verifies that these cells are not of hepatocytic lineage. Similarly, minimal detection of Kupffer cell or LSEC markers rules out contaminating macrophage or endothelial cell phenotypes, thereby solidifying the stellate cell profile.

Overall, these findings emphasize the usefulness of Col-GFP HSCs in liver research, particularly in the context of fibrosis, ECM remodeling, and stellate cell biology. The inclusion of GFP for real-time monitoring of collagen expression makes the cell line especially valuable for assays involving fibrogenic pathways and drug testing [4,12]. Depositing the cell line in a public biobank such as the DSMZ further ensures its accessibility to researchers worldwide, thereby fostering collaborative opportunities [40].

Moving forward, by carefully considering the chromosomal characteristics of this cell line and the occasional expression of atypical markers such as Keratin 19, investigators can design more robust experiments. Ultimately, Col-GFP HSCs are a powerful and accessible model that, when used alongside primary cell cultures and in vivo systems, can significantly enhance our understanding of hepatic fibrogenesis and stellate cell function.

It is important to note that Col-GFP HSCs are particularly well-suited for establishing robust 3D spheroid cultures, providing a more physiologically relevant environment compared to traditional two-dimensional systems [12]. These spheroids allow for more accurate modeling of tissue-like structures, reducing reliance on animal models and aiding in compliance with the 3R principle (Replace, Reduce, Refine). As a result, this cell line stands out as a valuable tool for advancing biomedical research in a more ethical and responsible manner.

Nevertheless, our data clearly demonstrate that the cell line does not fully replicate primary HSC biology. Specifically, the near-tetraploid karyotype and occasional chromosomal aberrations could influence certain signaling pathways. Although marker analysis strongly indicates an HSC origin, the moderate expression of cholangiocyte markers (e.g., keratin 19) and the lack of expression of some specific HSC markers at mRNA or protein levels suggest minor lineage deviations. Moreover, long-term culture adaptations in the cell line underscore the need for periodic verification of genomic integrity. Finally, the focus on C57BL/6 background cells may limit direct extrapolation to other mouse strains or human cells. Because of these limitations, any findings obtained in this immortalized cell line should be verified in primary HSCs. This ensures that the observed phenomena accurately reflect true HSC biology and account for potential differences arising from culture adaptation.

## 4. Materials and Methods

### 4.1. Literature Search

On 23 June 2025, a literature search was conducted using the search terms “Col-GFP-HSC” or “Col-GFP HSC” on PubMed [41] and the Google search engine [42]. The resulting hits were then screened for relevance to the cell line under investigation.

### 4.2. Cell Culture

Among the adherent and immortalized cell lines used, the mesenchymal-like Col-GFP HSC cell line was established in 2013 [4]. This cell line has been deposited with the DSMZ-German Collection of Microorganisms and Cell Cultures GmbH and is available there under catalogue number ACC 941. The hepatocyte-derived cell line AML12 was developed by the Fausto laboratory in 1994 [43]. AML12 hepatocytes were obtained from the American Type Culture Collection (#CRL-2254, ATCC, Manassas, VA, USA). Other immortalized hepatic stellate cells included in our study were the mouse cell line GRX (RRID: CVCL_M115) [44,45], rat cell lines CFSC-2G (RRID: CVCL_4U34) [46,47,48], HSC-T6 (RRID: CVCL_0315) [9,49], and PAV-1 (RRID: CVCL_7995) [50,51], as well as the human cell line LX-2 (RRID: CVCL_5792) [52,53]. Additionally, we used the cell line Rat-1 (RRID: CVCL_0492) [54,55]. Human SV40 large T-antigen transformed lung fibroblast cell line WI-38 [56,57] (RRID: CVCL_0579, clone VA-13, subline 2RA, #CCL-75.1) was obtained from ATCC (Manassas, VA, USA), and the human newborn foreskin fibroblast line NuFF (#GSC-3002) was obtained from GlobalStem (Rockville, MD, USA). All cell lines were routinely cultured in 10 cm dishes in a humidified incubator at 37 °C with 5% CO_2_ in a standard medium of Dulbecco’s Modified Eagle Medium (DMEM, high glucose, #D6171-500ML, Sigma–Aldrich, Merck, Taufkirchen, Germany), supplemented with 2 mM sodium pyruvate solution (#S8636-100ML, Sigma Aldrich), 10% fetal bovine serum (FBS, #F7524, Sigma–Aldrich), 2 mM L-glutamine (#G7513, Sigma–Aldrich) and 1× penicillin-streptomycin antibiotic solution (P0781-100ML, Sigma Life Science, Lonza, Cologne, Germany). For CFSC-2G, 1% non-essential amino acids (#11140-35, Gibco, ThermoFisher Scientific, Schwerte, Germany) were added to the standard medium. Cells were passaged when they reached 80–90% confluence using an Accutase solution (#A6964-100ML, Sigma–Aldrich) and seeded at the appropriate density for subsequent analyses.

### 4.3. Light Microscopic Analysis

For microscopic analysis, the cells were cultured to approximately 70–80% confluence and then briefly washed with phosphate-buffered saline (PBS). Live-cell morphology was observed using a Leica DM IL LED microscope equipped with a Leica EC3 camera and the Leica Application Suite (LAS) software (version 3.4.0), all from Leica (Wetzlar, Germany), with phase-contrast, bright field, or fluorescent illumination. Representative images were acquired at 100× or 200× magnification using a digital camera.

### 4.4. Electron Microscopy

For ultrastructural investigations, the Col-GFP HSC cells were collected and initially fixed in a solution of 2.5% glutaraldehyde prepared in a 0.1 M phosphate buffer. This was followed by post-fixation in a solution of 1% osmium tetroxide. After passing through a series of graded ethanol dehydration solutions, the samples were embedded in epoxy resin. Ultrathin sections, approximately 70–90 nm thick, were then prepared using an ultramicrotome and placed on copper grids. The sections were subsequently stained with uranyl acetate and lead citrate and examined with a Zeiss Leo 906 transmission electron microscope (Carl Zeiss AG, Oberkochen, Germany) operating at 60 kV. High-resolution micrographs taken at magnifications ranging from 2784× to 21,560× enabled a detailed evaluation of organelle preservation, including the structure of mitochondria and other notable subcellular features of Col-GFP HSC cells. Images were captured using a Zeiss Leo 906 transmission electron microscope operating at 60 kV.

### 4.5. Short Tandem Repeat Profiling

STR profiling of the murine cell lines Col-GFP HSC and AML12 was performed at IDEXX Laboratories in Kornwestheim, Germany, using the CellCheck™ Mouse 19 STR Profile and Interspecies Contamination Test. Genomic DNA was isolated from pelleted Col-GFP HSCs and AML12 cells and submitted to IDEXX for analysis. The CellCheck^TM^ Mouse 19 panel, which targets 19 mouse-specific STR loci, was used to verify the genetic authenticity and detect potential cross-species contamination. The results were provided in the form of an STR report, confirming the murine origin of both cell lines and providing a unique and expected fingerprint for each. The STR profiles of the control cell lines GRX, CFSC-2G, HSC-T6, PAV-1, and Rat-1 were previously established in our laboratory [45,46,49,51]. The authenticity of WI-38 cells was confirmed only by characteristic marker gene expression, including the SV40 large T-antigen [53].

### 4.6. RNA Extraction, Real-Time PCR, and Real-Time Quantitative PCR

The PureLink^TM^ RNA Mini Kit (#12183018A, Invitrogen, ThermoFisher Scientific) was used to extract total RNA from Col-GFP HSC and AML12 cells three days after seeding. As a positive control, RNA extracts from the liver of a 3-month-old adult male mouse collected in 2023 were used [58]. Tissue samples were homogenized using an MM400 mixer mill (Retsch GmbH, Haan, Germany) prior to RNA extraction. The RNA extraction process included DNase digestion using the PureLink^TM^ DNase set (#12185-010, ThermoFisher Scientific), following the manufacturer’s instructions. RNA concentration was determined spectrophotometrically. One µg of total RNA was reverse-transcribed into complementary DNA (cDNA).

For conventional RT-PCR, the cDNA underwent the following cycling conditions: Initial denaturation at 95 °C for 5 min, followed by repeated cycles of denaturation at 95 °C for 1 min, annealing for 1 min at either 60 °C for *Hnf4a* and *Rn18s*, 64 °C for *Acta2*, *Alb*, and *Gapdh*, or 59 °C for *Foxa2*, and extension/elongation at 72 °C for 3 min. This was followed by a final extension at 72 °C for 10 min. The amplicons were separated in 2% agarose gels containing 8 µL of Midori Green (#MG04, Nippon Genetics Europe, Düren, Germany), using a 1× TAE running buffer (40 mM Tris base, 20 mM acetic acid, and 1 mM ethylenediaminetetraacetic acid disodium salt dehydrate (EDTA)), and visualized using an iBright 750 imaging system (Invitrogen, ThermoFisher Scientific).

Quantitative PCR was conducted on a real-time PCR platform, following previously established protocols [58]. Gene expression was quantified and displayed relative to the housekeeping gene *Gapdh*. The 2^−ΔΔCt^ method was employed [59]. Each biological replicate was analyzed in triplicate to ensure consistent and reproducible data. Appendix A provides a comprehensive list of all the primers that were used.

### 4.7. Next-Generation Sequencing

Next-generation sequencing (NGS) analysis of the Col-GFP HSC cell line was carried out following established procedures [55]. High-quality RNA was isolated from plates of cells grown under basal conditions until they reached approximately 80% confluence. The cells were lysed and homogenized in a guanidine thiocyanate buffer, and the resulting lysate was layered onto a cesium chloride cushion. The RNA pellet was recovered, resuspended in sterile water, purified by ethanol precipitation, and then dissolved in sterile water. RNA integrity and concentration were evaluated using UV spectroscopy and confirmed with an Agilent 4200 TapeStation (Agilent Technologies Inc., Waldbronn, Germany). Subsequently, ribosomal RNA was depleted, and mRNA was reverse-transcribed for library preparation using the NEBNext Multiplex Oligos for Illumina Index Primers Set 1 kit. Sequencing was conducted on an Illumina MiSeq platform (Illumina Inc., San Diego, CA, USA) with pre-filled MiSeq Reagent kit V2, 300-cycle cartridges. The construction of the cDNA library, sequencing runs, and subsequent analysis were performed at the IZKF Genomic Facility, University Hospital Aachen.

### 4.8. Western Blot Analysis

Protein extraction from Col-GFP HSC and AML12 cells was performed 72 h after plating, following standard procedures [45]. The protein extracts (30 μg per lane) were heated at 80 °C for 10 min and then separated on 4–12% Bis-Tris NuPAGE^TM^ gels (#NP0335BOX, Invitrogen) under reducing conditions. The running buffer contained 50 mM 2-(*N*-morpholino)ethanesulfonic acid, 50 mM Tris-base, 0.1% sodium dodecyl sulfate, and 1 mM EDTA, at a pH of 7.2. After electrophoresis, the proteins were transferred to 0.45 μm nitrocellulose membranes (#10600002, Amersham^TM^ Protran^®^ Western-Blotting Membranes, Merck, Darmstadt, Germany) using 1x NuPAGE buffer. Transfer efficiency was confirmed with Ponceau S staining. The membranes were then blocked in Tris-buffered saline (50 mM Tris, 150 mM NaCl, pH 7.5) with 0.1% Tween 20 and 5% non-fat milk powder. Primary antibodies were applied, followed by horseradish peroxidase-conjugated secondary antibodies for detection using chemiluminescence (Supersignal™ West Dura Extended Duration Substrate, #34076, ThermoFisher Scientific). Liver protein extracts from a male mouse (80 μg per lane) collected in 2023 served as a positive control [45]. Tissue samples were homogenized using an MM400 mixer mill (Retsch GmbH). Appendix A provides a detailed list of all antibodies used.

### 4.9. Phalloidin Stain

40,000 Col-GFP HSC cells and 150,000 AML12 cells were seeded in a four-chamber polystyrene vessel tissue culture-treated glass slide (#354104, BD Falcon^TM^, BD Biosciences, Erembodegem, Belgium). A detailed step-by-step protocol with instructions can be found in [60]. After 24 h, the medium was removed, and the cells were washed three times with 1× PBS, then fixed in 4.0% paraformaldehyde (buffered with phosphoric acid to pH 7.4) for 20 min in the dark. Following fixation, the cells were washed three more times with 1× PBS and permeabilized with 0.2% Triton X-100 in PBS for 4 min on ice. After additional PBS washes, non-specific binding sites were blocked with 3% bovine serum albumin for one hour at room temperature. After further washing with 1× PBS, the cells were stained with a 1× diluted rhodamine-phalloidin solution (#R415, Invitrogen, ThermoFisher Scientific) for 20 min in the dark. Cell nuclei were then incubated with 4′,6-diamidino-2-phenylindole (DAPI, #D1306, ThermoFisher Scientific) for 30 min in the dark. This was followed by three washes with 1× PBS and one with dH_2_O. Finally, the samples were embedded in Vectashield Antifade mounting medium for fluorescence (#H-1000, Burlingame, CA, USA) and analyzed using a Nikon Eclipse E80i fluorescence microscope with NIS-Elements Vis software (version 3.22.01) from Nikon Europe (Düsseldorf, Germany).

### 4.10. Karyotype and M-FISH Analysis

For the combined karyotype and M-FISH analysis, Col-GFP HSCs were grown to approximately 70–80% confluence. The cells were then treated with colcemid (0.1 µg/mL) for 2–4 h to arrest them in metaphase. Previously published protocols for G-banding and M-FISH analysis were followed [54]. The cells were harvested by trypsinization and subjected to a hypotonic treatment in 0.56% KCl for 30 min at 37 °C. After hypotonic swelling, the cells were fixed in a 3:1 methanol/acetic acid solution and washed repeatedly until the supernatant was clear. Metaphase spreads were prepared by dropping the cell suspension onto cold, moist microscope slides and air-drying under controlled humidity. DAPI staining was used for banding to visualize individual chromosomes, identify structural abnormalities, and assess numerical changes. The 21XMouse—Multicolor FISH Probe for Mouse Chromosomes (MetaSystems, Altlussheim, Germany) was utilized for M-FISH analysis according to the manufacturer’s instructions to color-code each chromosome pair. Twenty-six metaphases were analyzed in detail using a Zeiss Axioplan fluorescence microscope (Carl Zeiss Jena, Jena, Germany), equipped with a CCD camera and ISIS image processing software (MetaSystems, Altlussheim, Germany) to establish the overall Col-GFP HSC karyotype.

## 5. Conclusions

Our comprehensive analysis shows that Col-GFP HSC cells exhibit the primary genetic, molecular, and morphological characteristics of activated HSCs. High-throughput sequencing, RT-qPCR, and Western blot analysis reveal the characteristic expression of fibrogenic markers and rule out significant expression of non-stellate liver cell markers. Electron microscopy confirms the expected ultrastructural features, and cytogenetic profiling reveals a stable karyotype with the typical alterations of an immortalized line. By depositing Col-GFP HSCs in a publicly accessible biobank, we are ensuring the widespread availability of this well-characterized, reproducible model system. Overall, these results validate the Col-GFP HSC line as an authentic HSC line, highlighting its potential for investigating stellate cell biology and the intricate mechanisms of liver fibrosis. Nevertheless, some limitations of this study should be acknowledged. First, we observed that certain classical HSC markers, including α-smooth muscle actin (*Acta2*) and glial fibrillary acidic protein (*Gfap*), were only weakly expressed at the transcript level. Although low levels of these markers are not uncommon in immortalized lines or under specific culture conditions, it remains possible that this partial downregulation may limit direct comparisons to fully activated primary HSCs. Furthermore, the near-tetraploid state and chromosomal rearrangements in Col-GFP HSC cells underscore that genetic adaptations to in vitro conditions can alter some endogenous signaling pathways or marker profiles. Future investigations using additional functional assays and side-by-side comparisons with freshly isolated primary HSCs may help clarify whether these alterations affect the translational relevance of the presented data.

## Figures and Tables

**Figure 1 ijms-26-07764-f001:**
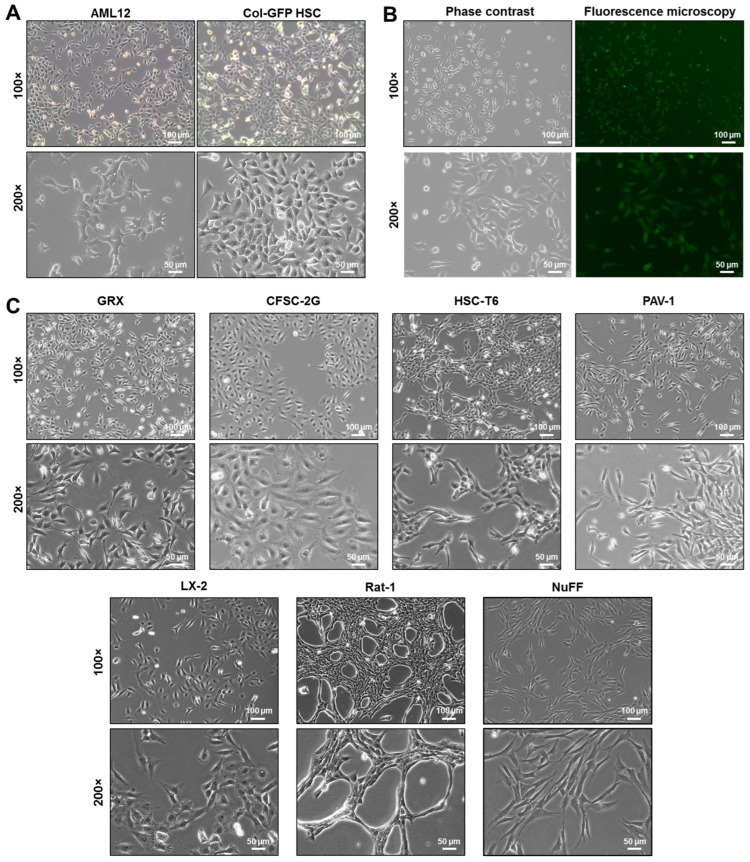
Microscopic analysis of the murine hepatic stellate cell line Col-GFP HSC and other immortalized cell lines: (**A**) Col-GFP HSC cells cultured in standard medium display an elongated, spindle-shaped, fibroblast-like morphology with a prominent nucleus. In contrast, hepatocytic AML12 cells form thin filopodia, resulting in a more irregular cell shape. (**B**) The Col-GFP HSC cell line contains a stably integrated transgene (pCol9GFP-HS4,5) that expresses green fluorescent protein (GFP) under the transcriptional control of the collagen type 1a1 promoter. This expression can be effectively monitored through fluorescence microscopy. (**C**) Microscopic features of other immortalized hepatic stellate cell and fibroblastic cell lines are shown, including hepatic stellate cell lines from mouse (GRX), rat (CFSC-2G, HSC-T6, and PAV-1), and human (LX-2), as well as a rat fibroblast cell line (Rat-1) and human neonatal foreskin fibroblasts (NuFF). Scale bars represent 100 µm for 100× magnification and 50 µm for 200× magnification.

**Figure 2 ijms-26-07764-f002:**
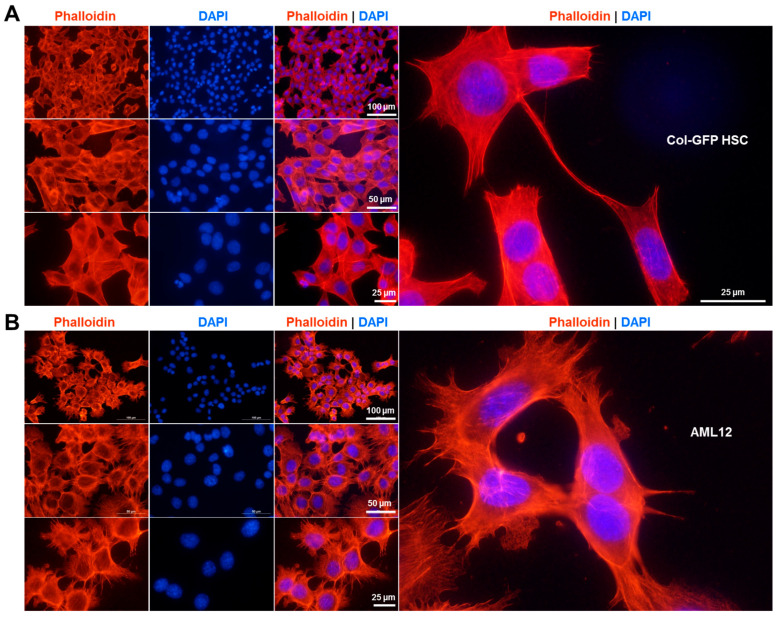
Rhodamine-phalloidin stain. Representative images of rhodamine-phalloidin and DAPI staining in (**A**) Col-GFP HSC and (**B**) AML12 cells are presented. The F-actin cytoskeleton is labeled by the rhodamine-phalloidin conjugate (red), while the nuclei are stained with DAPI (blue). Microscopic images were taken at 100× and 200× magnification, with scale bars indicating 100 µm, 50 µm, or 25 µm.

**Figure 3 ijms-26-07764-f003:**
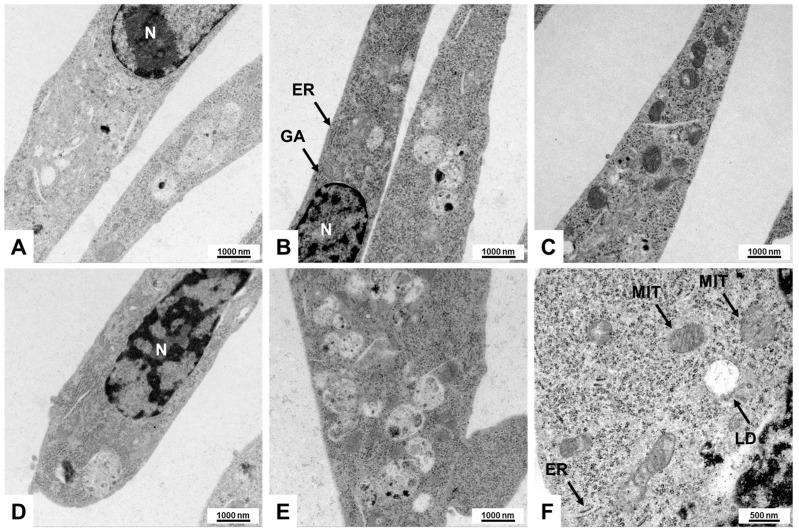
Electron microscopic features of Col-GFP HSCs: (**A**–**F**) Transmission electron microscopy reveals myofibroblast-like features, including abundant rough endoplasmic reticulum (ER), mitochondria (MIT), well-developed Golgi apparatus (GA), and occasional lipid droplets (LD), indicating retinoid storage capacity. The cells have a spindle-shaped phenotype, and the nuclei (N) are oval and euchromatic. These findings further validate the fibrogenic nature of Col-GFP HSCs at the subcellular level. Microscopic images were captured at 10,000× (**A**–**E**) and 21,560× (**F**) magnification, with scale bars representing 1000 nm or 500 nm.

**Figure 4 ijms-26-07764-f004:**
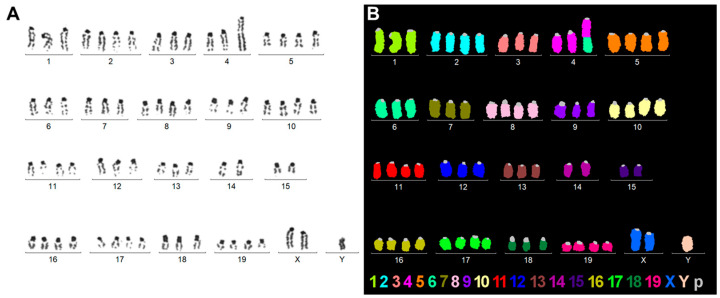
Karyotype and M-FISH analysis of Col-GFP HSC: (**A**) A representative inverted DAPI-banding-based karyotype and (**B**) an M-FISH result for the same metaphase, with colors assigned to each chromosome to confirm the C57BL/6 background. Most chromosomes were present in three or four copies, and the gonosomes were present as two X-chromosome and one Y-chromosome. Only one unbalanced translocation was observed, involving an exchange between chromosomes 4 and 6.

**Figure 5 ijms-26-07764-f005:**
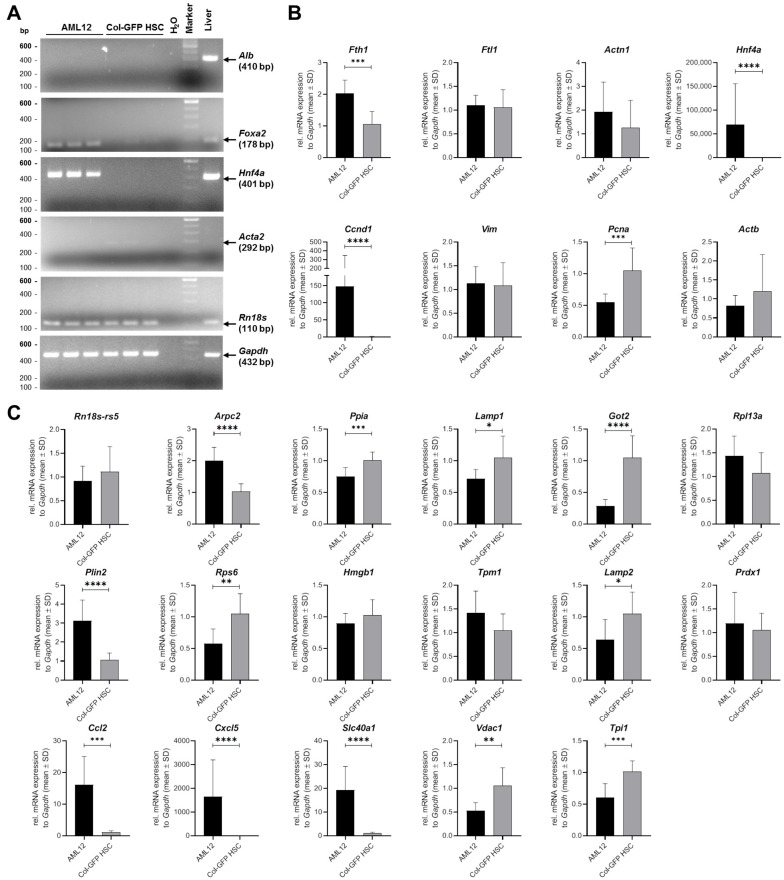
Hepatic stellate and hepatocyte marker gene expression in Col-GFP HSC and AML12 cell lines: (**A**) A representative standard RT-PCR analysis was conducted for the genes albumin (*Alb*), forkhead box A2 (*Foxa2*), hepatocyte nuclear factor 4-alpha (*Hnf4a*), alpha-2 smooth muscle actin (*Acta*), and the housekeeping genes 18S ribosomal RNA (*Rn18s*) and glyceraldehyde-3-phosphate dehydrogenase (*Gapdh*) in Col-GFP HSCs versus AML hepatocytes. Distinct band patterns illustrated the differential expression of genes between the two cell lines. (**B**,**C**) RT-qPCR measurements were performed for the indicated genes, with data expressed as a fold change normalized to *Gapdh*. The statistical analysis used either an unpaired *t*-test or a Mann–Whitney test. Significant differences between groups are indicated by asterisks (* *p* ≤ 0.05, ** *p* < 0.01, *** *p* < 0.001, **** *p* < 0.0001). The data are presented as mean ± SD.

**Figure 6 ijms-26-07764-f006:**
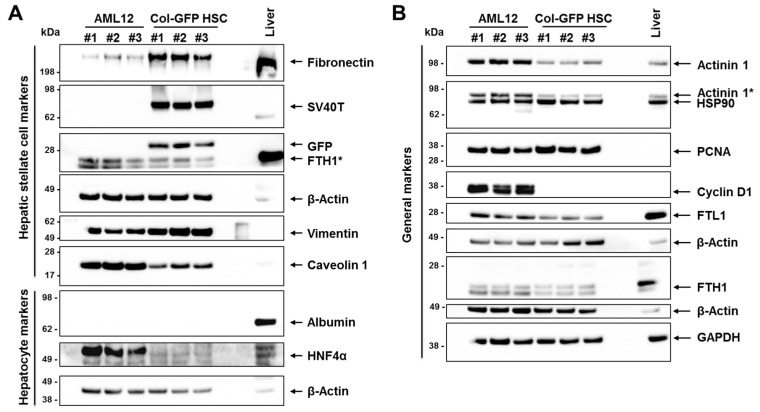
Western blot analysis: (**A**,**B**) Protein extracts were prepared from three independent cultures of each immortalized cell line (AML12 and Col-GFP HSC), with mouse liver tissue serving as a control. The protein extracts were then tested for specific marker proteins and probed with antibodies for SV40T antigen and green fluorescent protein (GFP). All blots were also probed with β-Actin or GAPDH to ensure equal protein loading. The symbol “#” refers to independent replicates, while “*” indicates a signal resulting from a previous probing. For more information, please refer to the text.

**Figure 7 ijms-26-07764-f007:**
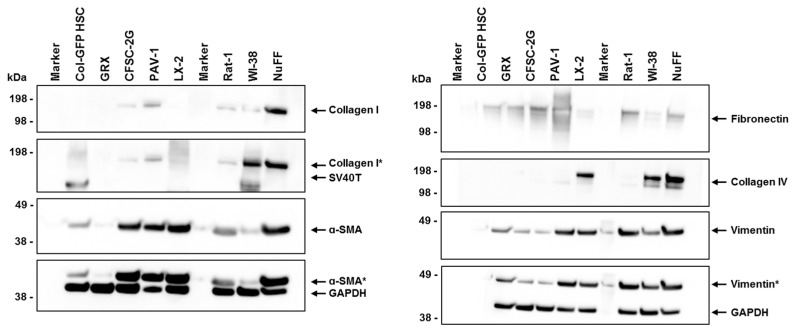
Expression of α-smooth muscle actin, fibronectin, vimentin, collagen type I, and collagen type IV in Col-GFP HSC, GRX, CFSC-2G, PAV-1, LX-2, Rat-1, WI-38, and NuFF cells. α-SMA, fibronectin, and vimentin are detected in all lines at varying intensities, while collagen type I is only expressed in CFSC-2G, PAV-1, Rat-1, WI-38, and NuFF. Collagen type IV is exclusively found in the human cell lines LX-2, WI-38, and NuFF. GAPDH expression was used as a loading control in this analysis. Some signals (highlighted in grey) are from previous hybridizations. When interpreting these data, it is crucial to consider that the antibodies used for α-SMA, fibronectin, collagen type IV, and vimentin are immunoreactive against mouse, rat, and human proteins. The antibody for collagen type I reacts with human protein (but can detect mouse and rat proteins), and the fibronectin antibody is reactive against mouse and rat proteins. Therefore, variations in signal intensities may be due to different affinities of the antibodies, making direct comparisons of signal intensities challenging. The symbol “*” indicates a signal resulting from a previous probing.

**Table 1 ijms-26-07764-t001:** Usage of Col-GFP HSCs in biomedical research.

Year of Publication	Major Findings	Reference
2013	This article introduces this mouse hepatic stellate cell line expressing green fluorescent protein (GFP) under the collagen 1(I) promoter and demonstrates its responsiveness to fibrogenic stimuli such as platelet-derived growth factor (PDGF) and transforming growth factor-β1 (TGF-β1). It also shows that endoglin overexpression enhances SMAD 1/5/8 and ERK1/2 signaling, upregulates fibrogenic markers, and reduces collagen I expression.	[4]
2018	The article found that p38 inhibitors increase the activation of other mitogen-activated protein kinases in various cell types, including Col-GFP HSCs.	[5]
2019	This study reveals that full-length endoglin (FL-Eng) localizes to caveolin-1-positive membrane domains and is incorporated into exosomes from hepatic stellate cells. It further demonstrated that all liver cell types can direct endoglin to exosomes, regardless of endogenous expression. *N*-glycosylation is not necessary for endoglin dimerization but is crucial for the secretion of soluble endoglin and the exosomal targeting of FL-Eng.	[6]
2021	The study examines the relevance of Perilipin 5 (PLIN5) in maintaining HSC quiescence in vivo and in vitro. Overexpression of PLIN5 suppresses the activation of the signal transducer and activator of transcription 3 (STAT3), as well as the TGF-β1-SMAD2/3 and SNAIL signaling pathways.	[7]
2022	A short tandem repeat (STR) profile for Col-GFP HSCs is established, enabling accurate authentication and helping resolve issues related to misidentification, cross-contamination, and genetic drift in biomedical research.	[8]
2022	The study reports the genetic characterization of the rat hepatic stellate cell line HSC-T6 using the Col-GFP HSC cell line as a control, which expresses typical HSC markers such as α-smooth muscle actin, collagen type I, fibronectin, and vimentin.	[9]
2023	This study provides information on the specific culture conditions and marker expression of Col-GFP HSCs and offers guidance on working with various other immortalized HSC cells from mice, rats, and humans.	[10]
2023	This communication summarizes available immortalized HSC cell lines from mice, rats, and humans, including Col-GFP HSCs.	[11]
2023	The study examines the effects of collagen modulatory compounds on GRX and Col-GFP HSC lines in (2D) monolayer and 3D mono-spheroid models.	[12]
2025	This article investigates aspects of mast cell biology, using Col-GFP HSCs as a control cell line for TGF-β signaling constituents (receptors and SMAD proteins) and responsiveness.	[13]

## Data Availability

The original contributions presented in the study are included in the article and Appendix A; further inquiries can be directed to the corresponding author.

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
