# Peer review of "Comprehensive Genetic and Molecular Characterization Confirms Hepatic Stellate Cell Origin of the Immortal Col-GFP HSC Line"

_ijms, 2025, doi:10.3390/ijms26167764_

Round 1

Reviewer 1 Report

Comments and Suggestions for Authors

Reviewer comments for: IJMS-3744801, Comprehensive Genetic and Molecular Characterization Confirms Hepatic Stellate Cell Origin of the Immortal Col-GFP HSC Line, by Buitkamp et al.

Summary

In their study, the authors characterize the immortalized Col-GFP HSC cell line. With that, the authors address an important aspect for preclinical research, the validity and the proper understanding of the cell lines used for in vitro research. The authors apply several methods, such as short tandem repeat profiling, karyotyping, RNA gene expression analysis, protein expression analysis, and morphological analysis to establish the characteristics of the Col-GFP-HSC cell line. From their analysis the authors conclude that the immortalized Col-GFP HSC cell line originates from mouse hepatic stellate cells without contaminations by other hepatic cell types, that the Col-GFP-HSC cell line retained features of hepatic stellate cells and consequently can serve as a reliable in vitro model for the study of hepatic stellate cell biology and pathobiology. The study adds important information to the community, however, there are several shortcomings that need to be addressed by the authors, as detailed below.

Major:

  1. In general, it remains questionable if the stated conclusions by the authors are sufficiently supported by the provided data and evidence. The authors propose that the Col-GFP HSC cell line retains a hepatic stellate cell phenotype (e.g. lines 168-169, 222, 227, and 275), however, the provided data to strengthen this claim is insufficient. No comparative data of the Col-GFP HSC cell line with other cells of mesenchymal origin are provided, which however, appears necessary to validate the conclusions made by the authors. The comparison to the AML12 (hepatocyte-derived) cell line is appreciated but does not provide the necessary specificity to establish the hepatic stellate cell phenotype of the Col-GFP HSC cell line. Without comparative data of the Col-GFP HSC cell line to other fibroblast-like cells (e,g, portal fibroblasts) the conclusions must be adjusted, and the relationship of the Col-GFP HSC cell line to other fibroblast-like cells of the mouse liver needs to be discussed.

  1. In their gene expression analysis (Table 2), the authors find no detectable (or minimal) expression of established hepatic stellate cell markers, such as Reln, Ngfr, Colec11, and Other markers, such as Lrat, Vipr1, and Pth1r are not included in the analysis. Instead, many genes commonly associated with fibroblast-like cells are, such as collagens are highly expressed (for reference, see e.g. https://doi.org/10.1016/j.celrep.2019.10.024). Since also fibroblasts are found in the mouse liver that natively express high levels of Col1a1 (and thus, also GFP in this model) it remains difficult to conclude a hepatic stellate cell origin of the Col-GFP HSC cell line from the gene expression analysis. Possibly, single-cell RNA-sequencing analysis of the Col-GFP HSC cell line might be recommended to obtain a better picture of possible contamination with other cell types and the possibility of comparison to transcriptomes of primary hepatic stellate cells (and other fibroblast-like cell types), available in many published studies.

  1. The authors perform several morphological analyses (phalloidin staining, electron microscopy) Col-GFP HSC cell line and conclude that observed features are consistent with hepatic stellate cell origin. However, the authors do not provide any comparative data to other cells of mesenchymal origin (e.g. another fibroblast-like cell line or primary isolated hepatic stellate cells), or liver tissue analysis that would strengthen their conclusions. Further, no references are provided to corroborate the conclusion made by the authors from their electron microscopy analysis (e.g. lines 222, 226-227, 286-287). At the least, references need to be provided to support the conclusion made by the authors.

Minor:

  • In Figure 1C, the phalloidin staining appears diffuse and no distinct, clearly visible stress fibers can be seen. Why is no phalloidin staining of the AML12 cell line provided?

Author Response

Dear Reviewer 1,

many thanks for reviewing our paper. Please find our comments to your suggestions in the attached PDF-File

Regards

Ralf Weiskirchen

Reviewer 2 Report

Comments and Suggestions for Authors

This manuscript provides a detailed and descriptive characterization of the Col-GFP hepatic stellate cell (HSC) line, confirming its hepatic stellate cell origin through genetic and molecular analyses. This work has the potential to significantly enhance the use of this cell line in research on HSC-related liver diseases, particularly in studies of liver fibrosis and related pathologies. My major concerns are:

  1. Comparison with Known HSC Cell Lines
    • The manuscript does not mention or compare the Col-GFP HSC line with other well-established hepatic stellate cell lines, such as LX-2, HSC-T6, or CFSC-2G, in the introduction or discussion.
    • A comparative analysis with these cell lines, particularly in terms of gene expression profiling, would provide context for the utility and novelty of the Col-GFP HSC line.
    • Ideally, these known cell lines should also be used as controls in the experiments, such as RT-qPCR or Western blot analyses, to validate the findings and highlight the unique features of the Col-GFP HSC line.
  2. Literature Search in the Results Section
    • Including a literature search within the results section is unconventional and disrupts the manuscript's flow.
    • This content should be moved to the introduction or discussion, where it can provide context for the study’s objectives or serve as a comparison with the findings.
  3. Flow of Data Presentation
    • The flow of data presentation is disjointed and appears to "jump" between different topics.
    • A more logical progression would improve readability. Specifically, I suggest the following sequence:
      • Start with cell morphology, then move to molecular-level analyses.
      • Place microscopic analysis (light/fluorescence microscopy) and electron microscopy next to each other for a cohesive presentation of structural data.
      • Follow with chromosome-level analyses (e.g., FISH).
      • Conclude with gene expression analyses (e.g., RT-qPCR and Western blot).
  4. Table 2 Placement
    • Table 2 does not provide critical data that is central to the manuscript’s main text. It would be more appropriate to move this table to the supplementary materials, where readers interested in the details can refer to it.
  5. Gene Expression Results for Section 2.4
    • The description of gene expression results in Section 2.4 is overly detailed and repetitive.
    • Since there is no comparison with other hepatic stellate cell lines, the current presentation lacks meaningful context.
    • Consider shortening this section, focusing on key findings, and emphasizing how these results confirm the hepatic stellate cell origin of the Col-GFP cell line.

Author Response

Dear Reviewer 2,

many thanks for reviewing our paper. Please find our comments to your suggestions in the attached PDF-File

Regards

Ralf Weiskirchen

Round 2

Reviewer 1 Report

Comments and Suggestions for Authors

Reviewer comments to ijms-3744801-v2

In the revised version of their manuscript, the authors have performed additional analysis, including the evaluation of other fibroblastic cell lines and complementary biochemical analysis. Thereby, and together with refined discussions within the text, the authors have strengthened their manuscript.

With their revision and the accompanying letter, the authors have addressed all previous comments.

Nevertheless, there are a few minor points to consider.

  • The statement about GFP expression as defining feature of the Col-GFP HSC line (line 98-99) may be reconsidered, since GFP expression is artificially induced in the line.
  • The annotation (cell lines at the top) of the right panel in Figure 7 appears slightly misplaced in relation.
  • The statement about ‘highly specific classical marker of HSC’ (line 398-401), may be reconsidered, since neither alphaSMA, vimentin, or fibronectin are specific markers for HSC. Which other suggested specific HSC markers were shown by the authors to be expressed in non-HSC cells?

Author Response

Reviewer 1

In the revised version of their manuscript, the authors have performed additional analysis, including the evaluation of other fibroblastic cell lines and complementary biochemical analysis. Thereby, and together with refined discussions within the text, the authors have strengthened their manuscript.

With their revision and the accompanying letter, the authors have addressed all previous comments.

Response: Thanks for re-evaluating our revised manuscript and the minor issues you pointed out. According to your suggestions, we have made the following changes.

Nevertheless, there are a few minor points to consider.

The statement about GFP expression as defining feature of the Col-GFP HSC line (line 98-99) may be reconsidered, since GFP expression is artificially induced in the line.

Response: Agreed. We have re-phrased the sentence to read: “Because the Col-GFP reporter is stably integrated into Col-GFP HSCs, GFP fluorescence provides a convenient, albeit artificial, read-out for collagen I promoter activity in this HSC line.”

The annotation (cell lines at the top) of the right panel in Figure 7 appears slightly misplaced in relation.

Response: We have corrected the layout in the source file and replaced Figure 7 with an updated version. The cell-line labels are now centered above the corresponding lanes.

The statement about ‘highly specific classical marker of HSC’ (line 398-401), may be reconsidered, since neither alphaSMA, vimentin, or fibronectin are specific markers for HSC. Which other suggested specific HSC markers were shown by the authors to be expressed in non-HSC cells?

Response: We agree that these markers are commonly used, but they are not specific to HSC. They can only help to distinguish HSCs from other resident liver cell types. Therefore, the paragraph has been modified. We have also added a new reference (Ref. 38) that discusses this in more detail and lined up all subsequent references.

Reviewer 2 Report

Comments and Suggestions for Authors

The authors addressed my concerns.

Author Response

Reviewer 2

The authors addressed my concerns.

Response: We appreciate the reviewer’s statement that all concerns have been addressed.